# Preparation and Performance of Nickel-Doped LaSrCoO_3_-SrCO_3_ Composite Materials for Alkaline Oxygen Evolution in Water Splitting

**DOI:** 10.3390/nano15030210

**Published:** 2025-01-28

**Authors:** Bangfeng Zong, Xiaojun Pan, Lifang Zhang, Bo Wei, Xiangxiong Feng, Miao Guo, Duanhao Cao, Feng Ye

**Affiliations:** 1School of Mechanical and Electronic Engineering, Suzhou University, Suzhou 234000, China; mailzong@163.com; 2Key Laboratory of Power Station Energy Transfer Conversion and System of MOE, School of Energy Power and Mechanical Engineering, North China Electric Power University, Beijing 102206, China

**Keywords:** nickel-doped, La_0.5_Sr_0.5_CoO_3_, sol-gel, oxygen evolution reaction, water electrolysis

## Abstract

Perovskites exhibit catalytic properties on the oxygen evolution reaction (OER) in water electrolysis. Elemental doping by specific preparation methods is a good strategy to obtain highly catalytical active perovskite catalysts. In this work, La_0.5_Sr_0.5_Co_1−x_Ni_x_O_3−δ_ perovskite materials doped with different ratios of nickel were successfully synthesized by the sol-gel method. The electrochemical measurement results show that for OER in 1 M KOH solution, La_0.5_Sr_0.5_Co_0.8_Ni_0.2_O_3−δ_ prepared by the sol-gel method requires only a low overpotential of 213 mV to reach 10 mA cm^−2^, which is significantly lower than that of La_0.5_Sr_0.5_Co_0.8_Ni_0.2_O_3−δ_ prepared by the hydrothermal method for the increasing about 45.24% (389 mV at 10 mA cm^−2^). In addition, La_0.5_Sr_0.5_Co_0.8_Ni_0.2_O_3−δ_ by the sol-gel method can be kept stable in an alkaline medium tested for 30 h without degradation. This indicates that the prepared La_0.5_Sr_0.5_Co_0.8_Ni_0.2_O_3−δ_ has better OER performance. The X-ray diffraction (XRD) results show that SrCO_3_ is the main phase formed, which is a disadvantage of this method. The performance improvement may be affected by the carbonate phase. The scanning electron microscopy (SEM) results show that layer structured La_0.5_Sr_0.5_Co_0.8_Ni_0.2_O_3−δ_ by the sol-gel method has more surface pores with a pore diameter of about 0.362 μm than spherical granular structured La_0.5_Sr_0.5_Co_0.8_Ni_0.2_O_3−δ_ by the hydrothermal method. X-ray photoelectronic spectroscopy (XPS) results reveal that the crystal lattice of La_0.5_Sr_0.5_Co_0.8_Ni_0.2_O_3−δ_ by nickel doping is lengthened, and the electronic configuration of Co is also changed by the sol-gel preparation process. The improved electrocatalytic performance of La_0.5_Sr_0.5_Co_0.8_Ni_0.2_O_3−δ_ may be attributed to the pore structure formed providing more active sites during the sol-gel process and the improved oxygen mobility with Ni doping by the sol-gel method. The doping strategy using the sol-gel method provides valuable insights for optimizing perovskite catalytic properties.

## 1. Introduction

Hydrogen energy is a highly efficient and clean source of green energy, widely utilized for energy storage and conversion, which aids in reducing reliance on fossil fuels [1]. However, the slow kinetics of the alkaline oxygen evolution reaction (OER) remain a major challenge, limiting the efficiency of hydrogen production from water electrolysis and hindering the industrial scalability of related technologies [2,3]. It is known that transition metal oxides, hydroxides (e.g., NiFeOH), and nickel-iron alloys exhibit good OER electrocatalytic performance, but their limited stability still restricts their use [4]. Therefore, developing new catalysts that combine high performance with durability is essential. Among potential alternatives, cobalt-based perovskites, such as PrCoO_3_, BaCoO_3_, and SrCoO_3_, have emerged as highly effective electrocatalysts for OER due to their favorable physicochemical properties, tunable structures, and cost-effectiveness [5,6]. In particular, recent studies highlight the advantages of cobalt-based perovskites in achieving superior catalytic performance for OER. For example, Zhao et al. [7] demonstrated that the perovskite Pr_0.7_Sr_0.3_Co_0.95_Ru_0.05_O_3_ exhibited a lower OER overpotential than the commercial RuO_2_ in 1 M KOH. Similarly, Ba_0.5_Sr_0.5_Co_0.8_Fe_0.2_O_3−δ_ was found to exhibit significantly better OER catalytic activity than iridium oxide catalysts [8]. These findings underscore the potential of cobalt-based perovskites as promising candidates for OER electrocatalysis in alkaline media.

Recent studies have demonstrated that doping and the incorporation of metal supports can significantly enhance the catalytic performance of perovskite materials [9]. For instance, combining Co_3_O_4_ with silver (Ag) or nickel (Ni) substrates has shown a synergistic effect, enhancing both oxygen evolution reaction (OER) and oxygen reduction reaction (ORR) activities [10]. Soltani et al. [11] demonstrated that metal-supported perovskite catalysts, such as Ag-supported Co_3_O_4_, exhibited significant improvements in electrocatalytic performance compared to bare metals or oxides, with the use of Ag as a substrate, reducing the overpotential of the OER by up to 40 mV. This enhanced performance can be attributed to several factors, including the higher electrical conductivity of the metal support, reduced contact resistance between the oxide and substrate, and electronic effects stemming from interactions between the metal and oxide components [12]. Additional studies have confirmed the potential of bimetallic systems. For instance, Hatem et al. [13] combined Ag with Co_3_O_4_ nanoparticles, demonstrating enhanced OER and ORR catalytic activity, with an ORR overpotential only 70 mV higher than that of commercial Pt catalysts. Moreover, metal-doped perovskites can significantly enhance OER activity by altering the electronic structure of the catalyst [14,15,16]. For example, Zhu et al. [17] synthesized a tetragonal structure of SrCo_0.95_P_0.05_O_3−δ_ by doping phosphorus into SrCoO_3−δ_. The synthesized SrCo_0.95_P_0.05_O_3−δ_ material achieved a current density of 10 mA cm^−2^ at a relatively small overpotential of 0.48 V. Luo et al. [18] synthesized Ba_0.9_Sr_0.1_Co_0.8_Fe_0.1_Ir_0.1_O_3−δ_, which demonstrated excellent OER activity under alkaline conditions. This enhancement is attributed to iridium substitution for iron, resulting in an overpotential of 300 mV at a current density of 10 mA cm^−2^, with stability maintained for 10 h in a 1.0 M potassium hydroxide electrolyte. It is noteworthy that among various dopants, nickel is the most effective candidate for enhancing the OER reaction [19]. Han et al. [20] synthesized nickel-doped perovskites (La_5_Ni_3_Co_2_) by optimizing the molar ratios of La, Ni, and Co. La_5_Ni_3_Co_2_ exhibited a current density of 10 mA cm^−2^ in 1.0 M potassium hydroxide at 0.360 V. These findings emphasize the importance of substrates and doping in optimizing the performance of perovskite-based catalysts. In contrast, there are relatively few studies on nickel-doped LaSrCoO_3−δ_ for alkaline OER, and further research on the catalytic performance of nickel-doped LaSrCoO_3−δ_ under various conditions is needed.

To date, various methods have been developed to synthesize these highly active perovskite materials, including electrostatic spinning, solution combustion, high-temperature methods, and hydrothermal techniques [21,22]. Niu et al. [23] prepared the La_0.6_Sr_0.4_Co_0.8_Ni_0.2_O_3−δ_ (LSCN-0.8) electrocatalyst using the electrostatic spinning method. LSCN-0.8 exhibited an overpotential of 327 mV in 1 M KOH at a current density of 10 mA cm^−2^. Saraswati et al. [24] synthesized La_0.6_Sr_0.4_Co_0.8_Ni_0.2_O_3_ using the solution combustion method, achieving a low overpotential of 250 mV at a current density of 10 mA cm^−2^. However, the perovskite materials prepared by the methods described above often suffer from grain agglomerations and inhomogeneity during the sintered process. The hydrothermal method has been particularly effective in achieving uniform particle sizes, resulting in perovskite materials with enhanced catalytic properties [25]. For example, the LaNiO_3_ electrocatalysts synthesized using the hydrothermal method exhibited a porous, hollow structure and remarkable stability, outperforming Pt/C catalysts in long-term stability tests [26]. In addition, perovskite materials can also be prepared using the sol-gel method. For example, Omari et al. [27] prepared LaNi_1−x_Co_x_O_3_ nanoparticles with a porous structure using the sol-gel method. Their optical band gap exceeded 3 eV, indicating that these perovskite materials are good semiconductors and catalysts. Liang et al. [28] synthesized impurity-free La_0.4_Sr_0.6_Co_0.8_Ni_0.2_O_3_ using the sol-gel method, exhibiting fewer agglomerates and a higher density of small particles and interstitials on the La_0.4_Sr_0.6_Co_0.8_Ni_0.2_O_3_ surface. This result indicates that high-purity, less agglomerated perovskite materials with controllable particle sizes can be successfully prepared via the sol-gel method. However, there are limited studies regarding the preparation of the LaSrCoO_3−δ_ system by the sol-gel method.

Herein, we synthesized nickel-doped perovskites La_0.5_Sr_0.5_Co_1−x_Ni_x_O_3−δ_ (x = 0.2, 0.5, 0.8) as electrocatalysts for the alkaline oxygen evolution reaction (OER) using the sol-gel method. The sol-gel method is employed to controllably synthesize highly homogeneous materials, which is anticipated to enhance the electrocatalytic performance of perovskite structures. As a comparison, a series of La_0.5_Sr_0.5_Co_1−x_Ni_x_O_3−δ_ (x = 0.2, 0.5, 0.8) was synthesized using the hydrothermal method. Electrochemical measurements were conducted to evaluate the OER properties of La_0.5_Sr_0.5_Co_1−x_Ni_x_O_3−δ_. Structural and morphological features were analyzed using various characterization techniques such as X-ray diffraction (XRD) and scanning electron microscopy (SEM). As the OER electrocatalyst, La_0.5_Sr_0.5_Co_1−x_Ni_x_O_3−δ_ synthesized via the sol-gel method was expected to exhibit good performance at certain ratios of cobalt/nickel by doping Ni, demonstrating a high current density, low overpotential, and stability in 1.0 M KOH electrolytes. Additionally, X-ray photoelectron spectroscopy (XPS) was used to investigate the relationship between the electrocatalytic properties of La_0.5_Sr_0.5_Co_1−x_Ni_x_O_3−δ_ by the sol-gel method with a specific ratio of cobalt/nickel and the change in the electronic configuration of Co. Furthermore, the reason for the good OER catalytic performance of the La_0.5_Sr_0.5_Co_1−x_Ni_x_O_3−δ_ catalyst was explored.

## 2. Experimental Section

La_0.5_Sr_0.5_Co_1−x_Ni_x_O_3−δ_ perovskite materials were prepared by the sol-gel method and the hydrothermal method, respectively. The raw materials, equipment, and calcination processes were the same except for the preparation methods. In addition, the ratios of nickel and cobalt were also consistent. The primary difference was the preparation process by the precursors.

### 2.1. Preparation of La_0.5_Sr_0.5_Co_1−x_Ni_x_O_3−δ_ Using the Sol-Gel Method

The molar ratio of cobalt and nickel was 10:1. Lanthanum nitrate (5 mmol), strontium nitrate (5 mmol), nickel nitrate (8 mmol), and cobalt nitrate (2 mmol) were dissolved in 80 mL of deionized water. Citric acid (5 mmol) was added, and the solution was ultrasonically dispersed for 10 min. After complete dispersion, a magnetic stir bar was placed in the beaker, which was then positioned on a magnetic heating stirrer rotating at 400 rpm, maintaining a heating temperature of 95 °C. The solution evaporated until it became viscous. Heating and stirring were stopped, and the stir bars were removed once the beaker cooled to room temperature. Subsequently, the beaker underwent further drying in a blast drying oven at 180 °C for 6 h. The dried material was then ground in a mortar and transferred to an alumina crucible, followed by placement in a muffle furnace. The furnace temperature was ramped from room temperature to 750 °C at 3 °C/min, held at 750 °C for 4 h, and naturally cooled to room temperature. This synthesis process was repeated for perovskite materials with x values of 0.5 and 0.8, maintaining consistent conditions, named as La_0.5_Sr_0.5_Co_0.5_Ni_0.5_O_3−δ_-S, La_0.5_Sr_0.5_Co_0.5_Ni_0.8_O_3−δ_-S, La_0.5_Sr_0.5_CoO_3−δ_-S, and La_0.5_Sr_0.5_Co_0.2_Ni_0.8_O_3−δ_-S.

### 2.2. Preparation of La_0.5_Sr_0.5_Co_1−x_Ni_x_O_3−δ_ by the Hydrothermal Method

In a beaker containing 60 mL of deionized water, 5 mmol each of lanthanum nitrate and strontium nitrate, 8 mmol of nickel nitrate, and 2 mmol of cobalt nitrate were dissolved. Then, 5 mmol of citric acid was added, and the solution was stirred continuously at 400 rpm for 2 h at room temperature. The resulting solution was transferred to a 100 mL Teflon hydrothermal reactor vessel and heated in an oven at 180 °C for 24 h. After cooling naturally, the product in the reactor liner was removed and washed with deionized water and ethanol using centrifugation. Once washing was complete, the product was placed in a blast drying oven at 110 °C for 12 h. The dried powder was then transferred to an alumina crucible, which was placed in a muffle furnace and heated at a rate of 3 °C/min until 750 °C. The powder was held at 750 °C for 4 h before cooling naturally to room temperature. The resulting material was La_0.5_Sr_0.5_Co_0.8_Ni_0.2_O_3−δ_. In the same preparation process, the cobalt-nickel ratio was adjusted to produce La_0.5_Sr_0.5_CoO_3−δ_-H, La_0.5_Sr_0.5_Co_0.5_Ni_0.5_O_3−δ_-H, La_0.5_Sr_0.5_Co_0.8_Ni_0.2_O_3−δ_-H, and La_0.5_Sr_0.5_Co_0.2_Ni_0.8_O_3−δ_-H.

### 2.3. Physical Characterization

Thermo Scientific^TM^ (Waltham, MA USA) Quattro S energy dispersive X-ray spectrum (EDS) and scanning electron microscopy (SEM) were used to determine the micromorphology and element content distribution of samples, respectively. A Panalytical X-ray diffractometer (Malvern, UK) equipped with an Al Kα radiation source (λ = 1.50) was used to get the X-ray diffraction (XRD) tests. A Thermo Scientific^TM^ ESCALAB 250Xi (Waltham, MA USA) spectrometer with a monochromatic Al Kα X-ray source (1486.6 eV) running at 200 W was used to perform the X-ray photoelectron spectroscopy (XPS). Prior to fitting the experimental peak with the XPSPEAK software (Version 4.1), all XPS peaks were calibrated using C 1s peak binding energy for adventitious carbon, which is 284.8 eV. All characterization experiments were performed at room temperature and atmospheric pressure.

### 2.4. Electrochemical Measurements

A typical three-electrode system in 1.0 M KOH electrolyte solution was used for all electrochemical testing in the electrochemical workstation (Bio-Logic VMP3, Seyssinet-Pariset, France). The electrolytes were saturated with N_2_ bubbles for 10 min before the electrochemical experiments. At a sweep rate of 5 mV s^−1^, the linear sweep voltammetry (LSV) curves were obtained for OER with 90% iR correction. The Tafel slopes were determined by fitting the overpotential-current density curves using the formula (η = a + b log∣j∣), where j, η, and b stand for the current density, overpotential, and Tafel slope, respectively. Electrochemical impedance spectroscopy (EIS) was performed with an amplitude of 5 mV at overpotentials of −0.1 V and 0.3 V between 100 kHz and 100 MHz.

## 3. Results and Discussion

### 3.1. Materials Characterization

The crystal structures of La_0.5_Sr_0.5_Co_1−x_Ni_x_O_3−δ_-S prepared by the sol-gel method were analyzed using XRD. As depicted in Figure 1, the diffraction peaks observed at 23.2°, 33.1°, 40.7°, 47.4°, 58.9°, 69.3°, and 78.9° in the four samples correspond to crystallographic planes of (012), (104), (202), (024), (300), (208), and (128) of La_0.5_Sr_0.5_CoO_2.91_ (PDF#48-0122). The diffraction peaks in the nickel-doped samples exhibit a leftward shift compared to the standard card, indicating that nickel ion doping has increased the lattice constant and expanded the crystal lattice spacing. Additionally, peaks at 25.2°, 25.8°, 36.5°, 44.1°, and 49.9° in the four samples correspond to SrCO_3_ (PDF#05-0418), with crystallographic planes identified as (111), (021), (130), (221), and (113), respectively. These peaks are significantly stronger than those of La_0.5_Sr_0.5_CoO_2.91_, suggesting that SrCO_3_ is formed as a main phase. A leftward shift of peaks described above demonstrates the coexistence of SrCO_3_ alongside La_0.5_Sr_0.5_CoO_2.91_ during the preparation process. As a result, the formation of SrCO_3_ is a drawback of the sol-gel method, which could be improved to reduce carbonate formation. Moreover, nickel-doped samples also exhibit small diffraction peaks at 43.0° and 62.7°. However, the La_0.5_Sr_0.5_CoO_2.91_ crystalline structure remains predominant in all samples, indicating that the perovskite structure formed after nickel doping resembles that of the pre-doped materials. Notably, the diffraction peak of La_0.5_Sr_0.5_Co_0.8_Ni_0.2_O_3−δ_ is more prominent in the prepared samples.

To obtain the morphologies of Sr_0.5_Co_0.8_Ni_0.2_O_3−δ_, Field emission scanning electron microscopy was used to analyze the morphology and surface element distribution of La_0.5_Sr_0.5_Co_0.8_Ni_0.2_O_3−δ_ prepared by the sol-gel method and hydrothermal method. Figure 2a–d displays the morphology of La_0.5_Sr_0.5_Co_1−x_Ni_x_O_3−δ_-S series materials synthesized using the sol-gel method at low magnification, showing a lumpy and lamellar structure across all materials. During oven drying, citric acid in the gel structure swells and combusts, producing carbon dioxide that escapes. This results in the formation of porous voids in the material. The structure of La_0.5_Sr_0.5_Co_1−x_Ni_x_O_3−δ_ consists of multiple laminated layers stacked on top of each other, with small debris on the surface without part of the main stack. Comparing materials with different nickel doping ratios, it is observed that La_0.5_Sr_0.5_Co_0.8_Ni_0.2_O_3−δ_-S and La_0.5_Sr_0.5_Co_0.5_Ni_0.5_O_3−δ_-S have similar surfaces at low magnification. However, most blocks have several irregularly distributed, more pronounced pores on their surfaces. The La_0.5_Sr_0.5_Co_0.2_Ni_0.8_O_3−δ_-S appears as broken blocks with few apparent holes, indicating gas escape during expansion and resulting in fragmentation.

Figure 3 illustrates the elemental distribution on the surface of La_0.5_Sr_0.5_Co_0.8_Ni_0.2_O_3−δ_-S. The elements La, Sr, Co, Ni, and O are evenly distributed on the surface. Some elements appear shaded or in darker colors, which mostly correspond to holes in the actual sample.

Figure 4 depicts the morphology of La_0.5_Sr_0.5_Co_1−x_Ni_x_O_3−δ_-H series materials synthesized using the hydrothermal method. These materials take on the shape of spherical particles. The overall volume is smaller than that of the blocks prepared by the sol-gel method; some particles are broken, and there is no apparent pore structure on their surfaces. Some particles are fragmented, and their surfaces lack apparent pore structures. A comparison of the materials with different cobalt-nickel doping ratios revealed that the particles of La_0.5_Sr_0.5_Co_0.8_Ni_0.2_O_3−δ_-H were significantly smaller than those with other ratios. The particle structures of all four samples are indistinguishable, with particles ranging in size from micrometers and covered with dendrites. La_0.5_Sr_0.5_Co_0.8_Ni_0.2_O_3−δ_-H is notable for its smaller particle size and dendrites measuring 50 to 60 nm upon closer examination.

The energy dispersive spectroscopy (EDS) of hydrothermally synthesized La_0.5_Sr_0.5_Co_0.8_Ni_0.2_O_3−δ_-H is shown in Figure 5. The sample surface exhibits a uniform distribution of five elements, with strontium being the least uniformly distributed among them. The prominent distribution of lanthanides suggests that a substantial fraction of these elements is concentrated on the surface of the sample.

### 3.2. XPS Analysis of La_0.5_Sr_0.5_Co_0.8_Ni_0.2_O_3−δ_ by the Sol-Gel Method

X-ray photoelectron spectroscopy (XPS) analysis of La_0.5_Sr_0.5_Co_0.8_Ni_0.2_O_3−δ_-S, prepared via the sol-gel method, shows significant shifts in the binding energies of La, Sr, Co, Ni, and O, indicating changes in the material’s electronic structure due to nickel doping. The binding energy peaks for La 3d, Sr 3d, Co 2p, Ni 2p, and O 1s, shown in Figure 6, were observed at characteristic values corresponding to their respective oxidation states, confirming the presence of La^3+^, Sr^2+^, and Co^3+^ in the samples. The La 3d peaks at 833.5 eV and 837.1 eV for the 3d_5/2_ orbitals and at 849.7 eV and 854.2 eV for the 3d_3/2_ orbitals, along with the Sr 3d peaks at 133.5 eV and 135.1 eV, matched the expected oxidation states [29,30]. Co 2p peaks at 779.6 eV and 794.9 eV indicate Co^3+^, with a shift in the Co 2p binding energy, suggesting an enhancement in Co oxidation state linked to improved electronic conductivity and catalytic activity [31]. Although the overlapping La peaks obscure the Ni 2p features, the positions for Ni^2+^ can be approximately identified [32]. The O 1s spectrum showed three distinct components: lattice oxygen (O^2−^) at 528.4 eV, surface-adsorbed oxygen at 531.6 eV, and CO_3_^2−^ at 532 eV, with the oxygen p-band center near the Fermi level [33]. Nickel substitution is thought to increase the B–O bond distance, which enhances both ionic and electronic conductivity by modifying Co electronic structure and shifting the diffraction peaks left [34,35]. These structural changes, including lattice extension, oxygen mobility enhancement, and electronic configuration modification, are consistent with previous reports [36,37] on doped perovskite-type oxides, where similar doping strategies improved catalytic performance. In conclusion, the synergistic effect of lattice extension and electronic modification improves the performance of La_0.5_Sr_0.5_Co_0.8_Ni_0.2_O_3−δ_-S.

### 3.3. Electrochemical Performance

The OER catalytic performance of La_0.5_Sr_0.5_Co_1−x_Ni_x_O_3−δ_-S was evaluated, as shown in Figure 7. Figure 7a shows that the LSV curves indicate that La_0.5_Sr_0.5_Co_0.8_Ni_0.2_O_3−δ_-S requires only 213 mV and 320 mV overpotentials to achieve current densities of 10 mA cm^−2^ and 50 mA cm^−2^, respectively. La_0.5_Sr_0.5_Co_0.8_Ni_0.2_O_3−δ_-S demonstrated excellent OER performance. Subsequently, La_0.5_Sr_0.5_CoO_3−δ_-S shows the next best performance, with current densities of 10 mA cm^−2^ at 397 mV overpotential and 50 mA cm^−2^ at 492 mV overpotential. La_0.5_Sr_0.5_Co_0.5_Ni_0.5_O_3−δ_-S and La_0.5_Sr_0.5_Co_0.2_Ni_0.8_O_3−δ_-S exhibited comparatively poor performance. The former exhibited an initial overpotential of 390 mV at 10 mA cm^−2^, but the current density did not increase proportionately thereafter. The latter displayed weak trends in both initial overpotential and the subsequent increase in current density, suggesting a potential deficiency in OER catalytic performance. Tafel plots were also used to analyze reaction kinetics. La_0.5_Sr_0.5_Co_0.8_Ni_0.2_O_3−δ_-S showed a Tafel slope of 180 mV dec^−1^, while La_0.5_Sr_0.5_CoO_3−δ_-S exhibited a lower slope of 150 mV dec^−1^. La_0.5_Sr_0.5_Co_0.5_Ni_0.5_O_3−δ_-S had a Tafel slope of 538 mV dec^−1^, indicating severely restricted reaction kinetics. In the impedance test, La_0.5_Sr_0.5_CoO_3−δ_-S, La_0.5_Sr_0.5_Co_0.8_Ni_0.2_O_3−δ_-S, and La_0.5_Sr_0.5_Co_0.5_Ni_0.5_O_3−δ_-S displayed similar electrical impedance values of 23.58 Ω,17.56 Ω, and 36.07 Ω, respectively. The fitting graph and relevant parameters of EIS are given in Appendix A (see Appendix A). In contrast, La_0.5_Sr_0.5_Co_0.2_Ni_0.8_O_3−δ_-S had a much higher impedance. This result mirrors the pattern observed in samples prepared via the hydrothermal method, suggesting that difficult electron transfer during catalysis significantly impacts the performance of the material. Additionally, as shown in Figure 7d, the overpotential of this type of material at a current density of 10 mA cm^−2^ is summarized by other researchers. This work has the lowest overpotential, proving that La_0.5_Sr_0.5_Co_0.8_Ni_0.2_O_3−δ_-S has excellent catalytic performance.

The OER catalytic performance of the La_0.5_Sr_0.5_Co_1−x_Ni_x_O_3−δ_-H was tested. The results are shown in Figure 8. The LSV performance curves reveal that La_0.5_Sr_0.5_Co_0.8_Ni_0.2_O_3−δ_-H exhibits good performance, with current densities of 10 mA cm^−2^ and 50 mA cm^−2^ and overpotentials of 389 mV and 490 mV, respectively. The Tafel value of La_0.5_Sr_0.5_Co_0.8_Ni_0.2_O_3−δ_-H is 179 mV dec^−1^. La_0.5_Sr_0.5_Co_0.2_Ni_0.8_O_3−δ_-H has the worst negligible OER performance. Overall, the catalytic performance through La_0.5_Sr_0.5_Co_1−x_Ni_x_O_3−δ_-H is much worse than that of La_0.5_Sr_0.5_Co_1−x_Ni_x_O_3−δ_-S, with a lower current density. The EIS results show that the catalytic performances of La_0.5_Sr_0.5_Co_0.5_O_3−δ_-H, La_0.5_Sr_0.5_Co_0.8_Ni_0.2_O_3−δ_-H, and La_0.5_Sr_0.5_Co_0.5_Ni_0.5_O_3−δ_-H have electrical impedance values of 32.22 Ω, 33.31 Ω, and 31.29 Ω, respectively, and the differences between them are minor. The fitting graph and relevant parameters of EIS are given in Appendix A and Appendix A. However, La_0.5_sr_0.5_Co_0.2_Ni_0.8_O_3−δ_-H also has a higher resistance, and the poor performance may be due to its extremely high electrical impedance, which hinders charge transfer during electrolysis.

La_0.5_Sr_0.5_Co_1−x_Ni_x_O_3−δ_-S was scanned at various speeds to generate cyclic voltammetry curves for each series and to plot the double-layer capacitance slope. According to Figure 9, La_0.5_Sr_0.5_Co_0.8_Ni_0.2_O_3−δ_-S exhibits the highest capacitance among the catalysts in this series, measuring 13.7 mF cm^−2^. La_0.5_Sr_0.5_Co_0.2_Ni_0.8_O_3−δ_-S follows closely with 8.2 mF cm^−2^. La_0.5_Sr_0.5_Co_0.5_Ni_0.5_O_3−δ_-S and La_0.5_Sr_0.5_CoO_3−δ_-S exhibit lower capacitances of 4.0 mF cm^−2^ and 1.7 mF cm^−2^, respectively. Combined SEM and double-layer capacitance analyses revealed that the morphology of the catalyst significantly affects the electrochemically active surface area, thereby influencing the OER activity.

During the cyclic voltammetry (CV) test, the cyclic voltammetry curves of the La_0.5_Sr_0.5_Co_1−x_Ni_x_O_3−δ_-H show remarkably low current densities. Despite its superior oxygen evolution performance, La_0.5_Sr_0.5_Co_0.8_Ni_0.2_O_3−δ_-H exhibits current densities less than one-tenth of those La_0.5_Sr_0.5_Co_0.8_Ni_0.2_O_3−δ_-S. The double-layer capacitance for La_0.5_Sr_0.5_Co_0.8_Ni_0.2_O_3−δ_-H is negligible according to CV calculations.

The comparison of SEM results from the two methods, using the double-layer capacitance test, revealed that the La_0.5_Sr_0.5_Co_0.8_Ni_0.2_O_3−δ_-S forms a porous structure during preparation. This improvement enhances the catalyst’s specific surface area, thereby increasing the number of active sites. In contrast, La_0.5_Sr_0.5_Co_0.8_Ni_0.2_O_3−δ_-H prepared by the hydrothermal method shows either complete or broken particle morphologies, with larger particle sizes and no change in specific surface area. Comprehensive analyses of bilayer capacitance tests indicate that the controllable preparation of the material morphology can have an impact on the catalytic activity of the material.

Stability testing of La_0.5_Sr_0.5_Co_0.8_Ni_0.2_O_3−δ_-S demonstrated the best OER catalytic performance. Nickel foam was selected as the substrate to prevent catalyst detachment and oxidation of the glassy carbon electrode surface during stability testing for oxygen evolution reactions. The catalyst was drop-coated onto the nickel foam, ensuring a consistent loading per unit area equivalent to that on the glassy carbon electrode during testing. Figure 10 shows that La_0.5_Sr_0.5_Co_0.8_Ni_0.2_O_3−δ_-S exhibits excellent catalytic stability even without pre-activation. Its potential remained stable, varying by approximately 10 mV over 30 h, demonstrating its robustness in alkaline conditions for OER.

## 4. Conclusions

In summary, nickel-doped perovskite oxides La_0.5_Sr_0.5_Co_1−x_Ni_x_O_3−δ_-S (x = 0.2, 0.5, 0.8) were synthesized using the sol-gel method. SEM and EDS results indicate that the La_0.5_Sr_0.5_Co_1−x_Ni_x_O_3−δ_-S presents numerous surface pores with a multilayered laminar structure and a homogeneous distribution of surface elements. La_0.5_Sr_0.5_Co_1−x_Ni_x_O_3−δ_-H prepared by the hydrothermal method shows spherical granular. La_0.5_Sr_0.5_Co_0.8_Ni_0.2_O_3−δ_-S requires only 213 mV and 320 mV overpotentials to achieve current densities of 10 mA cm^−2^ and 50 mA cm^−2^, respectively, for the oxygen evolution reaction (OER). La_0.5_Sr_0.5_Co_0.8_Ni_0.2_O_3−δ_-S shows excellent stability in alkaline solutions, and the potential change remains within about 10 mV after 30 h of stabilization tests. Comprehensive XRD and XPS analyses show that nickel doping extended the lattice and increased the B-O spacing, which changed the electronic configuration of cobalt and increased the oxygen mobility in the sol-gel process. As a result, the catalytic performance of the La_0.5_Sr_0.5_Co_0.8_Ni_0.2_O_3−δ_-S material for alkaline OER is improved. La_0.5_Sr_0.5_Co_0.8_Ni_0.2_O_3−δ_-S is a highly efficient, durable, and cost-effective electrocatalyst for water splitting in alkaline solutions. It offers valuable insights for improving the alkaline OER catalytic performance of non-precious metal perovskite materials in the future.

## Figures and Tables

**Figure 1 nanomaterials-15-00210-f001:**
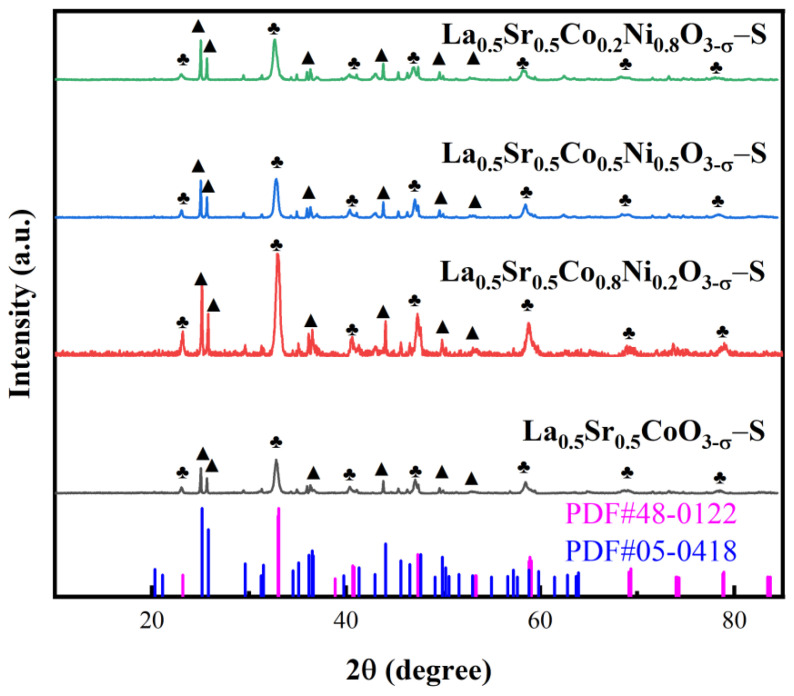
XRD images of La_0.5_Sr_0.5_Co_1−x_Ni_x_O_3−δ_-S by the sol-gel method.

**Figure 2 nanomaterials-15-00210-f002:**
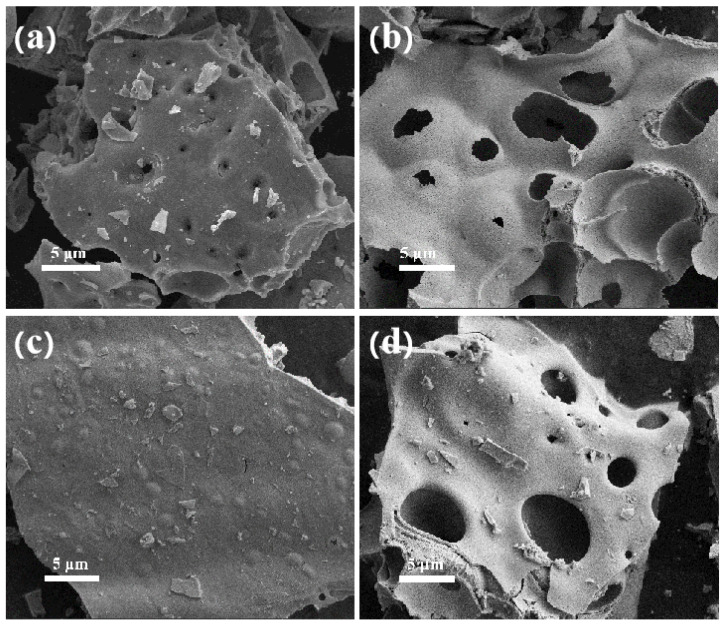
SEM images of (**a**) La_0.5_Sr_0.5_Co_0.8_Ni_0.2_O_3−δ_-S, (**b**,**c**) La_0.5_Sr_0.5_Co_0.2_Ni_0.8_O_3−δ_-S, and (**d**) La_0.5_Sr_0.5_CoO_3−δ_-S by the sol-gel method.

**Figure 3 nanomaterials-15-00210-f003:**
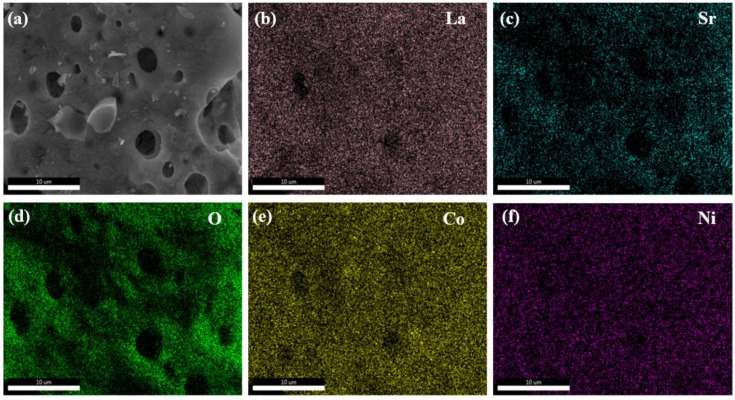
EDS mapping of La_0.5_Sr_0.5_Co_0.8_Ni_0.2_O_3−δ_-S by the sol-gel method at a 10 μm scale by SEM (**a**) SEM image of La_0.5_Sr_0.5_Co_0.8_Ni_0.2_O_3−δ_-S, (**b**) La, (**c**) Sr, (**d**) O, (**e**) Co, (**f**) Ni.

**Figure 4 nanomaterials-15-00210-f004:**
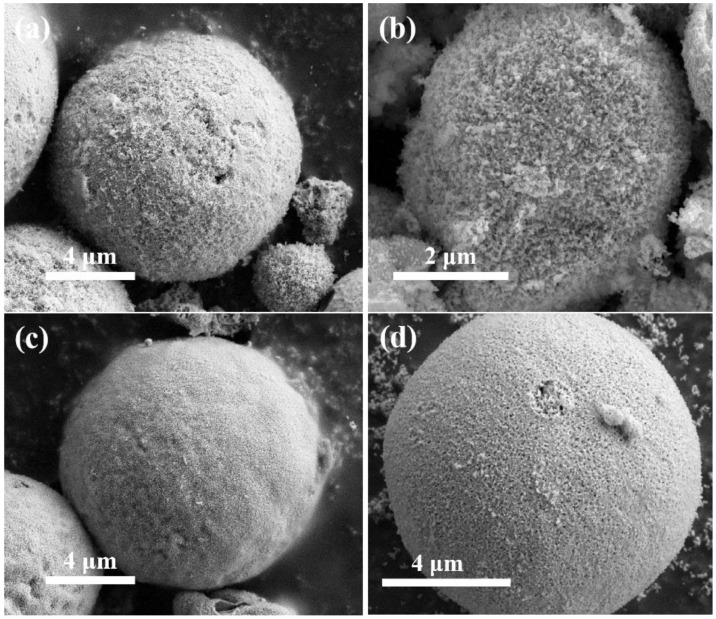
SEM images of (**a**) La_0.5_Sr_0.5_CoO_3−δ_-H, (**b**) La_0.5_Sr_0.5_Co_0.8_Ni_0.2_O_3−δ_-H, (**c**) La_0.5_Sr_0.5_Co_0.5_Ni_0.5_O_3−δ_-H, and (**d**) La_0.5_Sr_0.5_Co_0.2_Ni_0.8_O_3−δ_-H by the hydrothermal method.

**Figure 5 nanomaterials-15-00210-f005:**
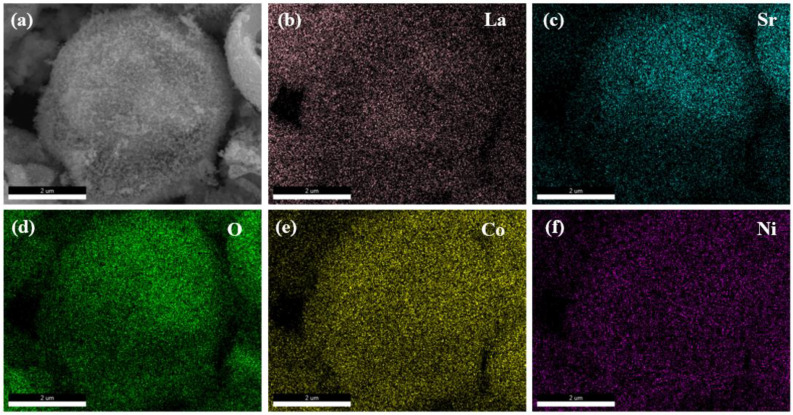
EDS mapping of La_0.5_Sr_0.5_Co_0.8_Ni_0.2_O_3−δ_-H by the hydrothermal method at a 2 μm scale by SEM (**a**) SEM image of La_0.5_Sr_0.5_Co_0.8_Ni_0.2_O_3−δ_-H, (**b**) La, (**c**) Sr, (**d**) O, (**e**) Co, (**f**) Ni.

**Figure 6 nanomaterials-15-00210-f006:**
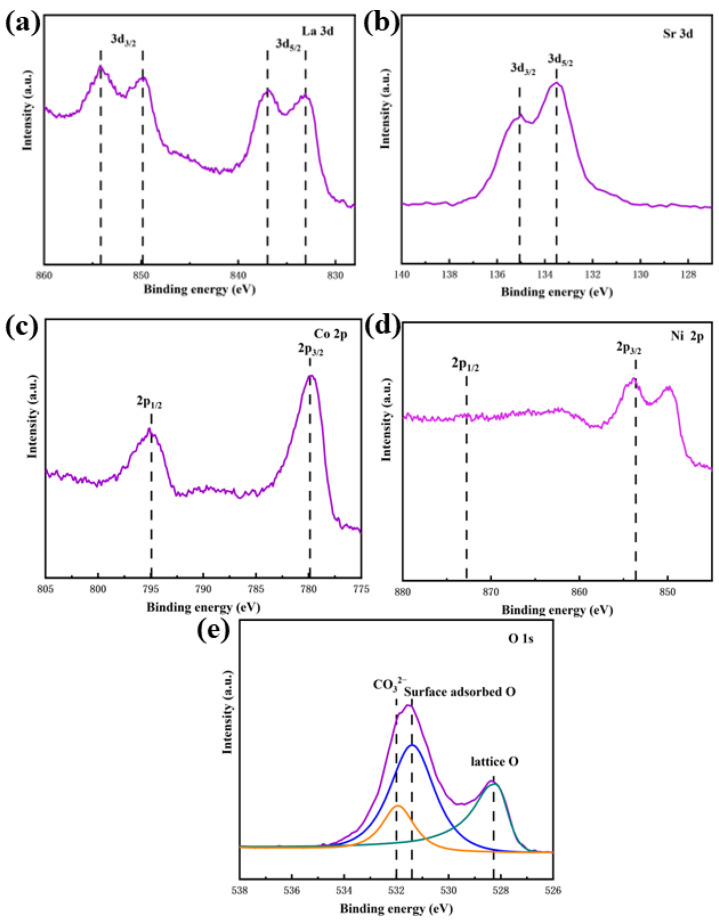
XPS spectra of La_0.5_Sr_0.5_Co_0.8_Ni_0.2_O_3−δ_-S by the sol-gel method. (**a**) La 3d, (**b**) Sr 3d, (**c**) Co 2p, (**d**) Ni 2p, and (**e**) O 1s.

**Figure 7 nanomaterials-15-00210-f007:**
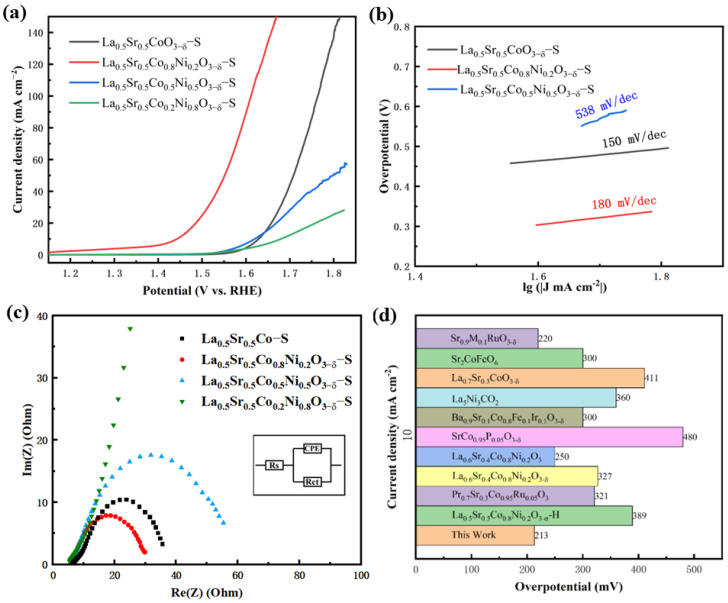
OER tests of La_0.5_Sr_0.5_Co_1−x_Ni_x_O_3−δ_-S (**a**) LSV curves, (**b**) Tafel curves, (**c**) EIS, (**d**) comparison of OER overpotentials of the catalysts in 1.0 M KOH solution at 10 mA cm^−2^ [7,8,17,20,23,24,38,39,40].

**Figure 8 nanomaterials-15-00210-f008:**
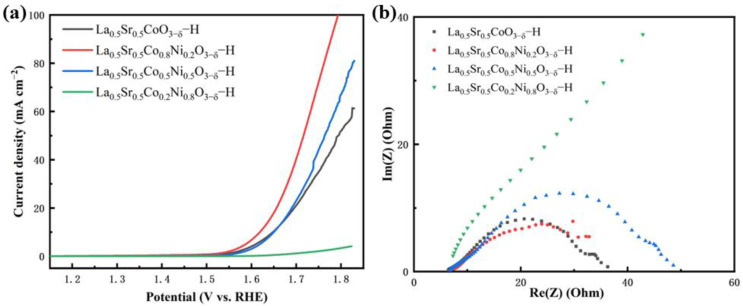
OER tests of La_0.5_Sr_0.5_Co_1−x_Ni_x_O_3−δ_-H. (**a**) LSV curves, (**b**) EIS.

**Figure 9 nanomaterials-15-00210-f009:**
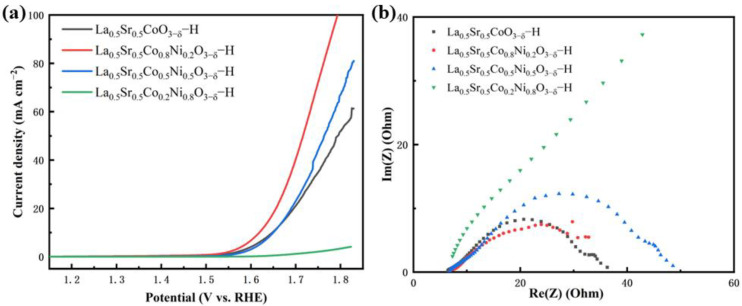
(**a**) CV curves of La_0.5_Sr_0.5_Co_0.8_Ni_0.2_O_3−δ_-S, (**b**) capacitive currents.

**Figure 10 nanomaterials-15-00210-f010:**
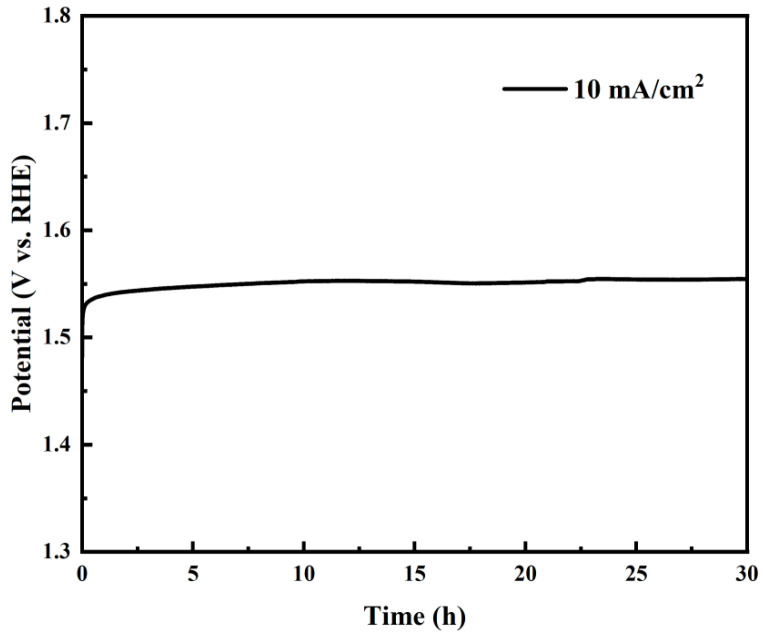
The stability test of La_0.5_Sr_0.5_Co_0.8_Ni_0.2_O_3−δ_ by the sol-gel method.

## Data Availability

The data presented in this study are available on request from the corresponding author.

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
