# Peer review of "Preparation and Performance of Nickel-Doped LaSrCoO3-SrCO3 Composite Materials for Alkaline Oxygen Evolution in Water Splitting"

_nanomaterials, 2025, doi:10.3390/nano15030210_

Round 1
Reviewer 1 Report
Comments and Suggestions for Authors
The submitted paper provides interesting results. Comparison of the OER performance of the nickel-doped La0.5Sr0.5CoO3-δ catalysts prepared by two different methods is well documented. However, some issues concerning the manuscript organization, references, and the discussion of the results should be addressed. The connection between the catalyst's structure and electrochemical performance should be better explained and supported by the literature.
1. Abstract - the first line – Write the acronym “… on the oxygen evolution reaction (OER)….”
- the fifth line – Specify the reaction: “The electrochemical measurement results show that for OER in 1 M KOH …”
- (389 @ 10 mA cm-2), correct to (389 mV @ 10 mA cm-2).
2. Introduction – the last paragraph – Techniques used for catalyst characterization should be only named in this part. Avoid prematurely giving results at this place, but rather emphasize the aim of this work and the expected advantage of the sol-gel synthesis.
3. Experimental – techniques used for materials characterization and electrochemical measurements, involving the equipment specification and the experimental conditions should be described (XRD, XPS, SEM, potentiostat, etc.).
4. Results – XPS characterization should be presented before the electrochemical results. There is no comparison of XPS data with the catalysts obtained by a hydrothermal procedure. References for each particular photoelectron line should be provided.
- Concerning this Section in general, there is a lack of discussion and relevant references.
5. Conclusions – Specify the reaction for which the overpotentials are given.
6. Author Contributions - missing
7. References – Almost all references are given in the Introduction. Expand the list of references that are important to discuss obtained results.
- Check the reference list. Journal names are inconsistent - write either the full names for all or abbreviated names for all.
Reviewer 2 Report
Comments and Suggestions for Authors
This manuscript describes the synthesis of La0.5Sr0.5Co1-xNixO3-δ-S (x = 0.2, 0.5, 0.8) using two different methods, sol-gel and hydrothermal. The materials were characterized and tested for the OER reaction in alkaline media. The catalysts showed different morphology and activity. Albeit several points explored by the authors are of great interest, some concerns have to be addressed before considering it for publication as follows:
1. Experimental details should be included: characterization methods details, was the catalyst activated/conditioned by cycling before recording LSV? EIS parameters? Were the LSVs iR-corrected?
2. The formed material isn’t phase pure as the peaks of SrCO3 are very strong and actually one can say you form this phase as a main phase, which is a draw back of the method. It would have been better to optimize the synthesis condition to avoid formation of carbonate. Please clearly and directly state this in the abstract and discussion.
3. The introduction should highlight the importance of developing noble-metal free catalysts especially cobalt-based perovskites and oxides for OER and aspect of generating synergistic and electronic effects by doping in the introduction by considering the following works (Metal-Supported Perovskite as an Efficient Bifunctional Electrocatalyst for Oxygen Reduction and Evolution: Substrate Effect), Boosting the bifunctional catalytic activity of Co3O4 on silver and nickel substrates for the alkaline oxygen evolution and reduction, and (A Highly Efficient Bifunctional Catalyst for Alkaline Air-Electrodes Based on a Ag and Co3O4 Hybrid: RRDE and Online DEMS.
4. Authors should hint to the importance of modulating lattice oxygen as they can participate in the reaction mechanism by referring to these works; How many surface atoms in Co3O4 take part in oxygen evolution? Isotope labeling together with differential electrochemical mass spectrometry, Role of Lattice Oxygen in the Oxygen Evolution Reaction on Co3O4: Isotope Exchange Determined Using a Small-Volume Differential Electrochemical Mass
5. Ni2+/3+ redox peak typically appears at 1.45V, however I can’t see this peak? Explain or magnify to show in the inset.
6. I doubt the reproducibility of the red curve in Figure 6a because changing the Ni from 0.2 to 0.5 has shifted the LSV by more than 150 mV which is too much, while the EIS of both is almost similar. Please repeat and double check.
7. XPS assignment of Ni 2p and Co 2p peaks can be supported by reference such as doi.org/10.1007/s12678-017-0364-z and 10.1016/j.mtchem.2023.101800
8. What are the given impedance values of 23.17 Ω,17.6 Ω, and 23.19 Ω? Authors should provide the equivalent circuit and the fitting parameters.
9. Authors should compare the activity of their perovskite with other relevant materials from literature
Round 2
Reviewer 1 Report
Comments and Suggestions for Authors
The authors have addressed all the issues satisfactorily. The manuscript is now acceptable for publication without further modification.
Author Response
Once again, I sincerely thank you for recognizing this work and for dedicating your valuable time and effort to it despite your busy schedule.
Reviewer 2 Report
Comments and Suggestions for Authors
Authors have made an effort to revise their manuscript and addressed most of the comments. However, to make it clear to the journal's community and readers, the following should be considered:
- Since the main phase of the synthesized material was found to be SrCO3 with some LaSrCoO3 perovskite, the title must be revised to reflect the actual studied material, maybe something like LaSrCoO3-SrCO3 material.
- It is important to include S-Fig1 and the discussion of the Ni2+/3+ in the manuscript or in a supporting information file to consolidate the work.
- If one assume that the LSVs in Figure 7a are reproducible which isn't trivial as such dramatic change emerges for only Ni0.2 and decreases to that extent for the Ni0.5, why do the EIS curves of Ni0.2 Ni0.5 in Fig 7c show almost the same Rct? the trend and changes of Rct should reflect the same trend of LSV.
- I don't think that the given Rct value of 17.6 Ohm matches the presented EIS curves in Fig. 7d. For proper analysis of data, please include the fitting lines in the curves and the fitting parameters in a table.
Note that the above comments from all reviewers and editor aren't aimed to postpone the publication process, but rather to ensure the high quality and reproducibility standards expected from the Editorial board and journals readers.
